# All-Solid-State Interdigitated Micro-Supercapacitors Based on Porous Gold Electrodes

**DOI:** 10.3390/s23020619

**Published:** 2023-01-05

**Authors:** Aymeric Pastre, Alexandre Boé, Nathalie Rolland, Rémy Bernard

**Affiliations:** 1PhLAM-Physique des Lasers Atomes et Molécules, CNRS, UMR 8523, Université de Lille, F-59655 Villeneuve d’Ascq, France; 2IEMN-Institut d’Electronique de Microélectronique et de Nanotechnologie, Université de Lille, CNRS, UMR 8520, F-59658 Villeneuve d’Ascq, France

**Keywords:** micro-supercapacitor, porous gold electrodes, electroless deposition, templating

## Abstract

Recent developments in embedded electronics require the development of micro sources of energy. In this paper, the fabrication of an on-chip interdigitated all-solid-state supercapacitor, using porous gold electrodes and a PVA/KOH quasisolid electrolyte, is demonstrated. The fabrication of the interdigitated porous gold electrode is performed using an original bottom-up approach. A templating method is used for porosity, using a wet chemistry process followed by microfabrication techniques. This paper reports the first example of an all-gold electrode micro-supercapacitor. The supercapacitor exhibits a specific capacitance equal to 0.28 mF·cm^−2^ and a specific energy of 0.14 mJ·cm^−2^. The capacitance value remains stable up to more than 8000 cycles.

## 1. Introduction

The current trend is for portable technology in the Internet of Things (IoT) to be increasingly smaller and smaller. Connected objects are based on wireless technology and need to be energetically autonomous [1]. Thus, the energy storage unit must be miniaturized as well. In addition, the energy consumption of connected objects is intermittent and highly variable. In this context, micro-supercapacitor (m-SC) technology appears as a promising candidate for future energy storage. Among all energy storage devices, m-SCs have attracted attention due to their high power density, high cycling stability, and their electrostatic storage [2,3,4]. They could offer an elegant solution to the problems of energy autonomy and power, while respecting the dimensional requirements of embedded systems. The ways of development of m-SCs are focused on the architecture, on the type of electrolyte, and on the nature of active electrode materials.

Interdigitated architecture is the most used configuration to scale down a supercapacitor, and shows two main advantages. First, interdigitated m-SCs exhibit dimensions compatible with an integration to classical electronic devices using microfabrication processes. In addition, the small gap between the electrodes in interdigitated m-SCs shortens the diffusion path of the ions, leading a to high specific energy [5,6,7].

Gel polymer electrolyte presents several advantages in terms of properties such as chemical and thermal stability, non-volatility, and wider electrochemical potential window, compared with liquid electrolyte. Moreover, with polymer gels it is possible to overcome the constraints inherent to the encapsulation of microdevices composed of liquid electrolytes [8,9], being a long and difficult process which is hard to integrate in classical microfabrication techniques.

Active materials for m-SCs electrodes are mainly made of metal oxides, nanostructured carbon or conductive polymers [10,11,12,13]. However, metal, and especially gold, appears to be a promising candidate as a supercapacitor electrode material due to its high electrical conductivity and chemical stability [5]. As a matter of fact, gold is already involved in microfabrication processes, and particularly in the manufacture of planar micro- supercapacitors, but only as a current collector. The small specific surface is the main issue with the use of such material for electrodes. Creating porosity within gold films makes it possible to increase specific surface and fabricate small devices with high performance (high surface-area-to-volume ratio). Three-dimensional nanoporous gold presents the advantage of being used as both current collectors and electrodes [14]. Most of these devices are fabricated using a dealloying technique or template-assisted electrochemical deposition.

Few examples of supercapacitors based on porous gold electrodes are described in the literature. The fabrication of mesoporous all-metal electrodes was first reported in 2010 [15]. Such material was obtained by a top-down dealloying process, which consists of the desorption of weak organic ligands of gold and silver nanoparticles, followed by a chemical etching of silver particles. Only one example of an on-chip interdigitated supercapacitor containing porous gold is reported in literature. The gold was used as a porous electrode, combined with manganese oxide as an active material, and with PVA/H_2_SO_4_ as the electrolyte.

In this paper, the fabrication of interdigit-patterned electrodes is reported. The manufacturing process leads to an all-integrated micro-supercapacitor on silicon wafer, including a quasisolid Polyvinyl Alcohol/Potassium hydroxide (PVA/KOH) electrolyte. Electrodes are fabricated through a bottom-up approach, using templating and wet chemistry processes. Polystyrene microspheres were used in this study as a template, because this material allows tailoring of the pores, is easily auto-assembled and is easily removable. The electrical and structural properties of the films are investigated and demonstrate the regularity of the porous film and its capacitive behavior. The integrated micro-supercapacitor is characterized in terms of its electrochemical properties and aging behavior.

The paper is organized as follows. The fabrication of the m-SC is reported in Section 2. Section 3 describes the results in terms of the physical properties of the porous gold electrodes and the electrochemical performance of the integrated micro-supercapacitor.

## 2. Materials and Methods

This section describes the manufacturing of the porous gold thin film material used as an electrode to realize an integrated micro-supercapacitor.

Zirconium (IV) n-propoxide (Zr(OC_3_H_7_)_4_) at 70 wt.% in n-propanol, 2-propanol (over molecular sieve, >99.5%), acetylacetone (>99%), tetrachloroauric acid (hAuCl_4_·3H_2_O, >99.9%), sodium borohydride (NaBH_4_, >96%), hydrogen peroxide solution (H_2_O_2_, 30 wt.% in H_2_O), hexane (>95%) and ethanol (>99.8%) were purchased from Sigma-Aldrich (France), and used without any further purification. Milli-Q water with a resistivity of 18.2 MΩ cm was used in all the preparations. The silicon substrates used were 3.5 × 2 cm² P-doped (5–10 Ω·cm) samples of Si (100) wafers.

The fabrication process and characterization of the porous gold films by an original “bottom-up” approach, combining polystyrene microsphere templates and gold electroless deposition method, has been thoroughly presented in a previous work [16]. Briefly, the process consists of alternating the deposition of polystyrene microsphere assembly and gold thin film electroless deposition. The subsequent removal of the polystyrene template leads to a gold film with a tailored and interconnected porosity. This film is anchored to the silicon substrate thanks to an Au/ZrO_2_ adhesion/seed layer.

The steps of the fabrication process are summarized in Figure 1, and are detailed in Section 3.

## 3. Results and Discussion

### 3.1. Fabrication of the Porous Gold-Based Micro-Supercapacitor

The fabrication process of the porous gold-based micro-supercapacitor includes six main steps, compatible with the classical microfabrication processes (see Figure 1), and these are detailed in the following subsections. First, an Au/ZrO_2_ seed layer is deposited to promote the adhesion of the gold thin film on the substrate (steps 1 & 2) [17]. Second, a mold is fabricated to pattern the gold film using a standard photolithography process (step 3). Then the gold thin film-embedding polystyrene microspheres are deposited using an electroless method (step 4) [18]. After microsphere and photoresist removal, the electrical contacts of the m-SC are deposited and patterned (step 5). Finally, the electrolyte is deposited and dried to finalize the micro-supercapacitor (step 6).

As a first step, the silicon substrate is cleaned by soaking in a 1:1 (H_2_SO_4_, 96%: H_2_O_2_, 30 wt.%) piranha solution for 10 min at 110 °C. Then, the substrate is deoxidized into a hydrofluoric acid (HF, 1 wt.%) for a duration of 1 min. An oxide layer is thermally grown through a wet oxidation process at 1100 °C. A 200 nm-thick layer ensures the electrical insulation of the substrate. The oxidized substrate is again decontaminated by soaking in a piranha solution, and activated by an oxygen (O_2_) plasma treatment in a PlasmaLab80 (Oxford) PECVD (Plasma Enhanced Chemical Vapor Deposition) chamber (Figure 1a). The treatment allows the creation of silanol groups (Si–OH) on the surface of the SiO_2_ layer when the ultraclean surface comes in contact with the air humidity [19,20,21]. The prepared substrates can be stored in milliQ^®^ water until they are used within a period of one month [22].

The gold nanoparticles embedded in a zirconia matrix (Au/ZrO_2_) were synthesized using a sol–gel process described in previous reports [17]. A dip-drawing technique is used to deposit the Au/ZrO_2_ seed layer. The time of immersion into the Au/ZrO_2_ stock solution (10 s), the drawing speed (100 mm·min^−1^) and the drying time (2 h at room temperature) are optimized in order to obtain a 300 nm-thick, crackless film; see Figure 1b.

The silicon substrate coated with the Au/ZrO_2_ adhesion/seed layer is dehydrated on a hot plate at 110 °C under air for 10 min, in order to promote the adhesion of the photoresist. A negative AzNlof2020 photoresist is deposited on the substrate using the spin-coating technique, before it is dried on a hot plate at 110 °C under air for 1 min. The photoresist is then selectively exposed to a 365 nm UV beam (MA6 Alignment, Karl Suss, SUSS MicroTec, Garching, Germany) for 3.7 s at 10 mW·cm^−2^ through a photomask presenting interdigitated comb patterns (cf. Figure 2). Then, the photoresist undergoes an inversion annealing step on a hot plate at 110 °C in air for 1 min. Finally, the photoresist is developed for 50 s in a pure MIF726 developer solution. The quality of the pattern is verified by optical microscopy. The photoresist thickness, measured by contact profilometry, is about 1.9 μm. This thickness allows the growth of a porous gold film up to 1 µm.

At the end of the photolithography process, the sample reveals a selective part of the Au/ZrO_2_ adhesion/seed layer with an interdigitated comb structure (Figure 1c). This uncoated part is where the porous gold film will be grown. The adhesion/seed layer is successively coated with a gold film, an assembly of polystyrene microspheres (template), and a gold film covering the template. The PS microspheres were synthesized by a modified microemulsion polymerization method inspired by a previous study [18] with diameters ranging from 20 to 100 nm. Several iterations of the template/gold thin film deposition steps lead to a Au/PS multilayered film (Figure 1d). The polystyrene template is dissolved by soaking in toluene, and the remaining photoresist layer is removed, revealing the three-dimensional interdigitated comb structure (Figure 1e). The dissolution of the photoresist is carried out by soaking in a Remover PG [23] bath at 70 °C for 30 min. The sample is cleaned with isopropyl alcohol. Finally, rectangular-shaped ohmic contacts connected to the electrodes by metallization lines are deposited using a lift-off method (Figure 1e). These ohmic contacts are exclusively used for the electrochemical characterization of the microdevices.

The gel polymer is prepared according to the following conditions: 3 g of polyvinyl alcohol (PVA) is dissolved in 30 mL of milliQ^®^ water. The solution is heated to 85 °C, stirred for 2 h at 1200 rpm, cooled to room temperature, and stirred again for one hour. Then, 2.24 g of potassium hydroxide (KOH) is dissolved in 10 mL of milliQ^®^ water, and the KOH solution (1 M) is added to the aqueous PVA solution. The PVA-KOH solution is stored in a round-bottom flask, sealed with a glass stopper until it is used. This solution can be kept under these conditions only for a few days. The PVA-KOH solution is therefore systematically freshly prepared before each deposition to avoid any aging of the polymer.

The gel polymer electrolyte is deposited using a dip-drawing technique. The sample is vertically immersed in the PVA-KOH solution and a primary vacuum is applied to the system for two hours. The sample is then removed from the solution at a constant rate of 50 mm/min. The ohmic contacts are cleaned with a lint-free absorbent paper previously soaked in water. Finally, a primary vacuum is applied to the sample for 12 h in order to extract residual water and to achieve the drying of the polymer film (Figure 1f), according to Zeng et al.’s work [24].

### 3.2. Interdigitated Comb Patterns

The characteristic dimensions of the interdigitated patterns are shown in the following figure. The term w represents the finger width (μm), i the interdistance between the electrodes (μm), L the finger length (μm) and the term n represents the number of fingers. These dimensions are included in the reference pattern in the general form: w_a_i_b_L_c_n_d_. A schematic view of the interdigitated structure with the design parameters can be seen in Figure 2.

According to previous work on carbon-based microelectrodes carried out by D. Pech et al. in 2013 [7], the dimensions of the interdigitated patterns (w, i, L, n) play an important role in optimizing the electrochemical performance of the micro-supercapacitor. In this study, measurements performed on gold current collectors with various interdigitated patterns in aqueous H_2_SO_4_ (0.5 M) allowed the authors to establish the dependence of the electrolyte resistance and the current collector resistance with respect to the configuration of the microdevice. They concluded that a decrease of the interdistance, i, leads to a decrease of the contribution of the electrolyte resistance. This results in an enhancement of the specific power of the microdevice. Additionally, they established that the resistance of the current collector increases with the enlargement of the active surface of the electrode, and with the increase of the parameters w, L, n and the decrease of the parameter i. This results in an enhancement of the specific energy of the microdevice. In order to obtain a compromise between a large specific energy and a high specific power, the pattern w_100_i_5_L_1000_n_10_ was chosen for this study.

The interdigitated w_100_i_5_L_1000_n_10_ microelectrodes were characterized using scanning electron microscopy (SEM). The SEM images (Figure 3) clearly exhibit the deposition of the electrode material in the form of interdigitated combs, separated by a 5 µm-wide gap (Figure 3a). In addition, an enlargement of part of Figure 3a highlights the good quality of the photolithography process, showing sharp edges and vertical walls, with low distortion patterns (Figure 3b). The 5 μm resolution obtained with the fabrication process described in this work is comparable to other results previously reported in the literature [25]. Experiments conducted to attain 1 μm resolution showed an absence of structuration of the patterns.

The process that was developed offers the advantage of simultaneously depositing many patterns on the same substrate (collective fabrication), as illustrated in Figure 3c.

### 3.3. Study of Gel Polymer Electrolyte Incorporation

Experiments using various polymer concentrations from 2.5 wt.% to 12.5 wt.% were performed. When the PVA concentration is higher than 8 wt.%, the dip-drawn gelled polymer film, after the drying step, exhibits complete delamination, as can be seen in Figure 4a. Polymer films prepared using lower concentrations and deposited using the dip-drawing technique exhibit no delamination after drying, as shown in Figure 4b. The process that was developed has the advantage of simultaneously coating all the patterns deposited on the substrate, as illustrated in Figure 4c.

Cross-sectional views of the final device were performed using scanning electron microscopy. Figure 5a shows that the polymer electrolyte film was uniformly coated on the surface of the nanoporous gold film. The PVA/KOH film has an average thickness of 3 μm. A zoomed-in view was obtained of a part of the nanoporous gold film after incorporation and drying of the polymer electrolyte (Figure 5b). This SEM image clearly shows the presence of PVA/KOH in the bulk of the nanoporous gold film. This confirms the penetration of the gelled polymer electrolyte into the entire porous volume of the microelectrode material.

### 3.4. Electrochemical Characterizations of the Devices

The electrochemical performance of the micro-supercapacitor was characterized using a SP-150 galvanostat from Biologic for cyclic voltammetry. All the characterizations were performed using a two-electrode configuration under air. The scan rate was varied from 1 mV·s^−1^ up to 500 mV·s^−1^.

Figure 6a illustrates the behavior of the m-SC for two scan rates (respectively 10 mV·s^−1^ and 100 mV·s^−1^). Figure 6b summarizes the evolution of the specific capacitance with respect to the scan rate. The shape of the curves is typical of electrostatic micro-supercapacitors. The Nyquist plot shown in Figure 6c is also characteristic of the behavior of a micro-supercapacitor.

The specific capacitance was assessed using the following equation:Cspecific=idvdtS
where i is the mean current, dvdt the scan rate, and S the active surface of the electrodes. The specific energy density was calculated by the following equation:$$E_{Specific}=\frac{CV^{2}}{2S}$$
where C is the specific capacitance of the material, and V the potential window for discharging.

The specific capacitance is equal to 284 µF·cm^−2^ at 10 mV·s^−1^ and 246 µF·cm^−2^ at 100 mV·s^−1^. At 500 mV·s^−1^, the capacitor behavior is maintained with a specific capacitance equal to 236 µF·cm^−2^, but without a square shape, showing the impact of the series resistor and thereby the RC constant effect. The specific energy calculated at 10 mV·s^−1^ is equal to 142 μJ·cm^−2^.

The electrochemical stability of the m-SC was investigated using cyclic voltammetry (charging and discharging) at 20 mV·s^−1^. The evolution of the specific capacitance is shown in Figure 7. The inset shows the initial cycle, the 1000th, and the 5000th cycle. The capacitance slightly increases during the first 1000 cycles; this is probably due to better impregnation of the electrolyte (almost 8 wt.% more in specific capacitance at the 1000th cycle compared with the initial one). The specific capacitance slightly decreases thereafter, showing the ageing of the gel electrolyte. After 8500 cycles at 20 mV·s^−1^, the specific capacitance is still higher than 95% of its initial value, demonstrating the stability of the m-SC and the very good stability of the capacitive behavior of the electrodes.

Comparable with other gold-based microdevices, an example of an interdigitated supercapacitor using porous gold as current collector with manganese oxide as the active material exhibits a specific capacitance equal to 30 µF·cm^−2^ [24]. More recently, a flexible interdigitated micro-supercapacitor with gold nanowires as the electrode material was reported. This device exhibits a 60.8 μF·cm^−2^ specific capacitance at a scan rate of 20 mV·s^−1^, decreasing to 13.3 μF·cm^−2^, with a capacitance retention of 21.9%, at a higher scan rate of 10 V·s^−1^ [26].

## 4. Conclusions

An integrated micro-supercapacitor was successfully designed and fabricated on a silicon substrate using microfabrication technology. The supercapacitor is made of porous gold electrodes, patterned as interdigitated comb structures using the lift-off technique. The porous gold electrodes were fabricated using a stack of electroless deposition of gold and polystyrene microspheres, leading to a porous gold film after the polystyrene was removed. Adhesion to the silicon oxide on top of the silicon substrate is ensured by a sol–gel deposited Au/ZrO_2_ layer, also acting as a seed layer. A quasisolid PVA/KOH electrolyte was used. The integrated micro-supercapacitors were characterized using SEM and cyclic voltammetry at different scan rates from 1 mV·s^−1^ up to 500 mV·s^−1^. The capacitive behavior of the porous gold electrodes is demonstrated at all scan rates. For a 1 M KOH concentration, the specific capacitance at 10 mV·s^−1^ is equal to 0.28 mF·cm^−2^ and the specific energy is equal to 0.14 mJ·cm^−2^. After 8500 charging-discharging cycles, the capacity is higher than 95 % of initial capacity, demonstrating the extremely stable capacitive behavior of the electrodes.

## Figures and Tables

**Figure 1 sensors-23-00619-f001:**
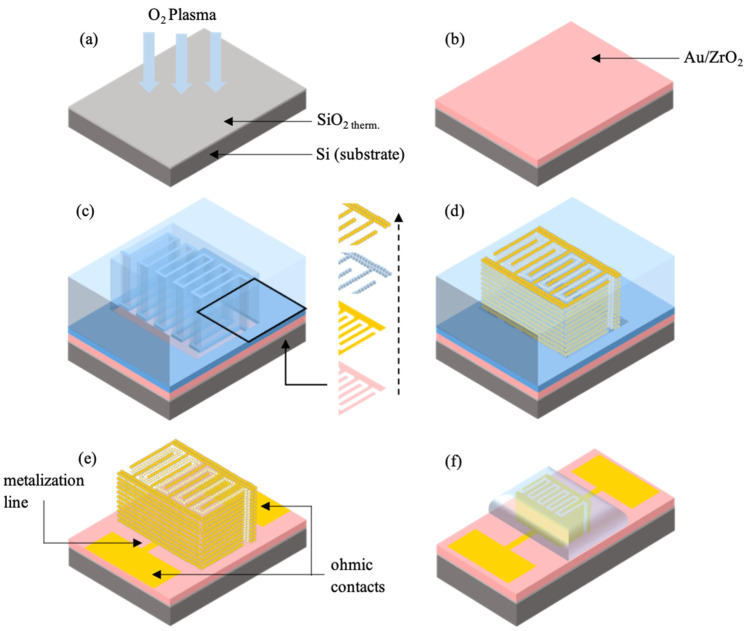
Schematic diagram of the porous gold-based micro-supercapacitor fabrication process. (**a**) Substrate functionalization by O_2_ plasma treatment; (**b**) Au/ZrO_2_ adhesion/seed layer deposited by dip-drawing; (**c**) photolithography process and porous gold film growth (gold electroless plating, template deposition and template metallization); (**d**) Au/PS multilayered film deposited by dip-drawing, (**e**) photoresist removal and electric contact deposited by sputtering; (**f**) gel polymer electrolyte incorporated by dip-drawing and by applying vacuum.

**Figure 2 sensors-23-00619-f002:**
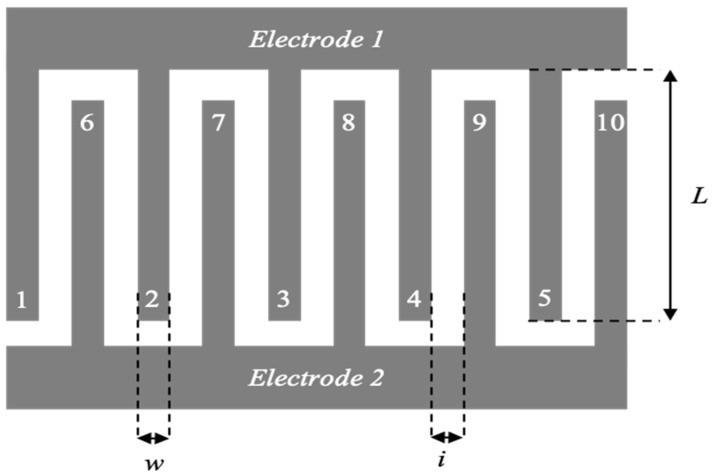
Schematic representation of the reference pattern w_100_i_5_L_1000_n_10_.

**Figure 3 sensors-23-00619-f003:**
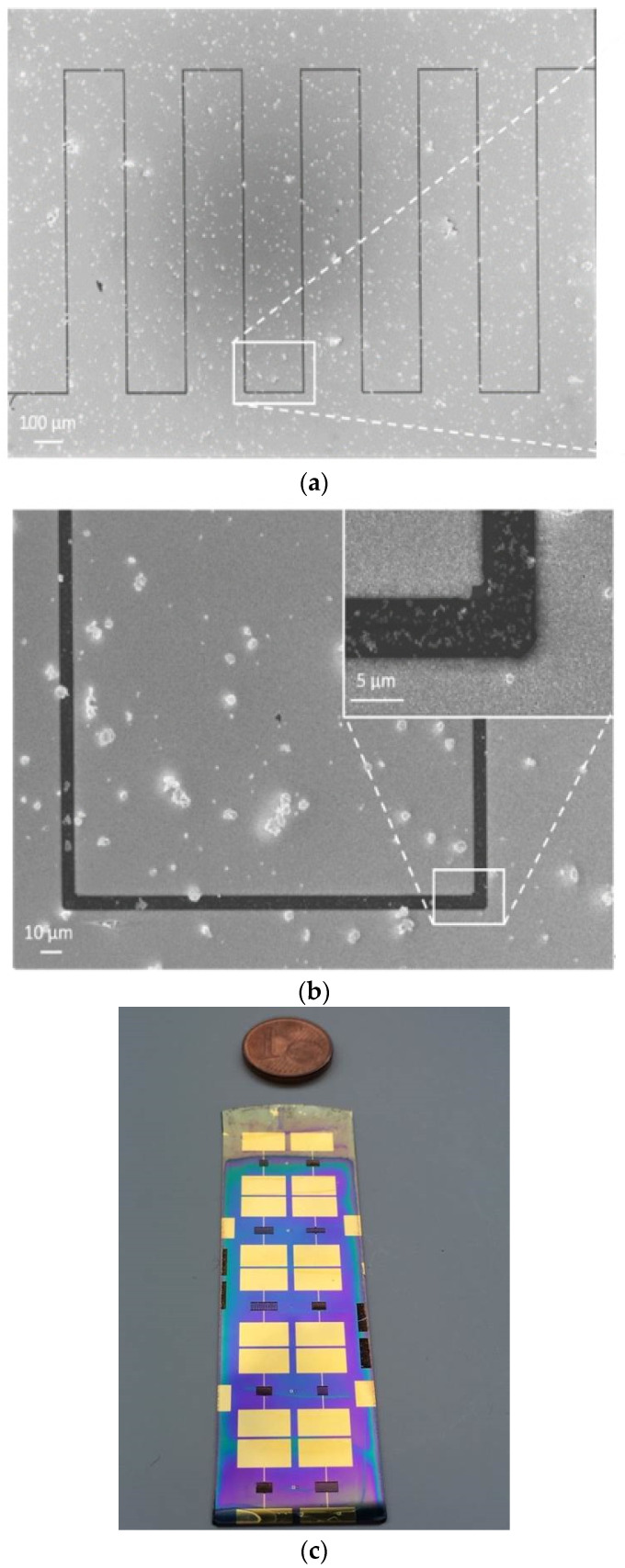
SEM images of the reference sample (w_100_i_5_L_1000_n_10_) viewed from above with magnification (**a**) ×148 and (**b**) ×1000. The inset represents a zoomed-in view on the angle of a comb. (**c**) Photograph of a piece of silicon substrate containing ten interdigitated microelectrodes and their ohmic contacts.

**Figure 4 sensors-23-00619-f004:**
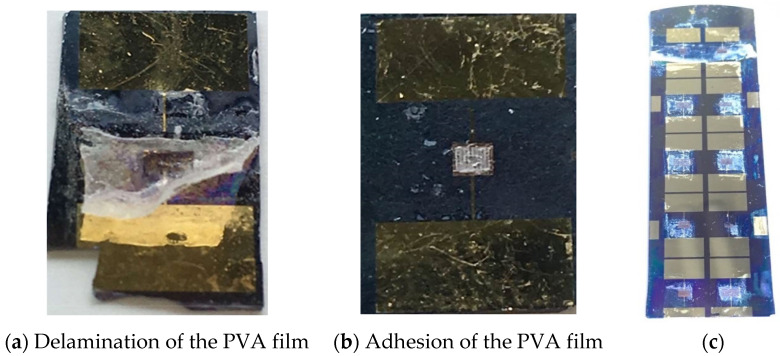
Photographs of microdevices fabricated from (**a**) 10 wt.% aqueous (delamination) and (**b**) 7.5 wt.% PVA solutions. (**c**) Photograph of 10 microdevices fabricated on the same substrate, top view.

**Figure 5 sensors-23-00619-f005:**
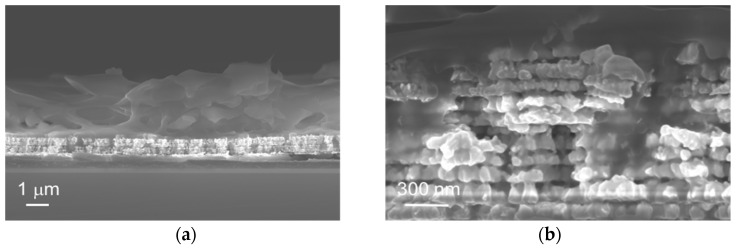
SEM images of a nanoporous gold film after incorporation and drying of PVA/KOH, cross-sectional view: (**a**) image taken with magnification ×20,000, (**b**) zoomed-in view of the porous gold film.

**Figure 6 sensors-23-00619-f006:**
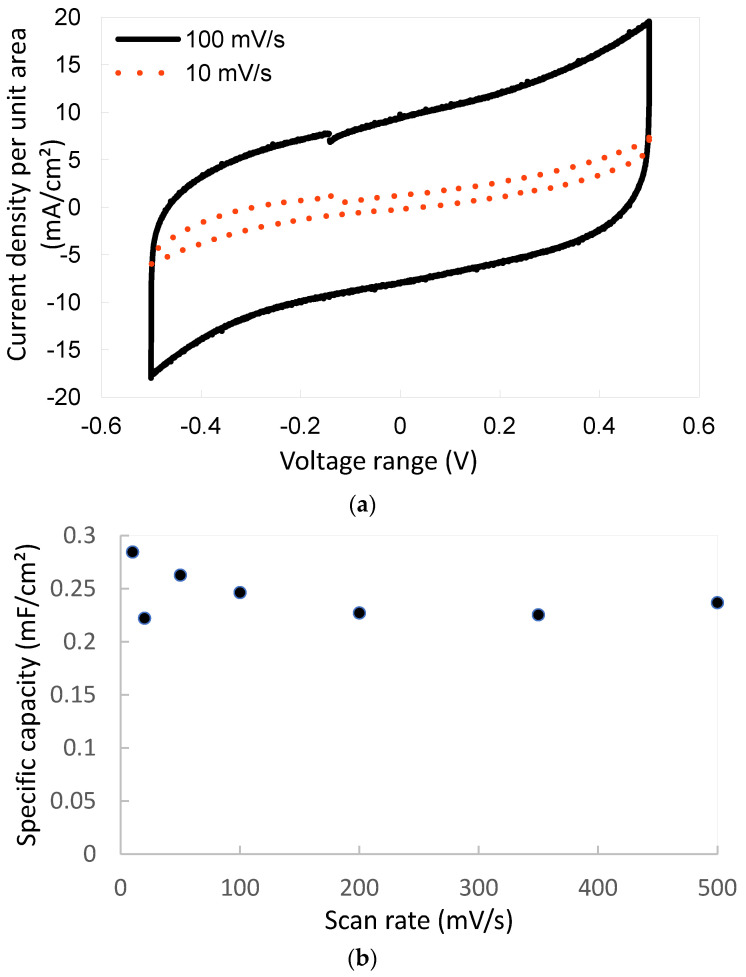
(**a**) Galvanostic cycling of the m-SC at a scan rate of 10 mV·s^−1^ and 100 mV·s^−1^ (**b**) Evolution of the specific capacitance vs. the scan rate, and (**c**) Nyquist plot.

**Figure 7 sensors-23-00619-f007:**
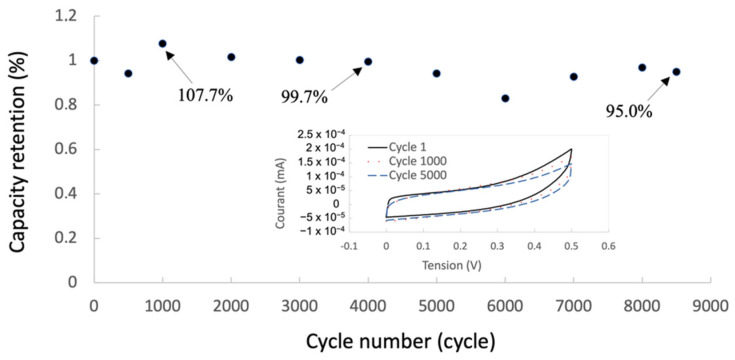
Evolution of the specific capacitance as a function of cycles at 20 mV·s^−1^. The inset shows voltammograms for the initial, 1000th and 5000th cycles.

## Data Availability

No new data were created or analyzed in this study. Data sharing is not applicable to this article.

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
