# Peer review of "All-Solid-State Interdigitated Micro-Supercapacitors Based on Porous Gold Electrodes"

_sensors, 2023, doi:10.3390/s23020619_

Round 1

Reviewer 1 Report

The current article demonstrates the electrochemical applications of gold-induced solid state micro supercapacitor.

The article has significant data to be published in this journal. Therefore, I recommend acceptance of this article after "Minor" revision. My comments are given below -

1. Please check the unit of specific capacity in Fig. 6(b).

2. Fig. 6(b): It is well known the the capacity tends to decrease with increasing the scan rate. But, herein te capacity was initially decreased and then increased. The authors should explain this anomalus behavior. 

3.  Nyquist plot should be given

4. The authors should calculate the energy density of the device.

5. Ragone plot of the device should be given.

6. A comparison of the electrochemical performance of the device with other related published devices should be provided. 

Reviewer 2 Report

The authors prepared interdigitated all gold-based micro-supercapacitor with relatively good electrochemical performance. The concept and design of the study are excellent, and the data presented is also perfect. Here are some suggestions for improvements:

1.      The Au/ZnO2 materials synthesis needs to be described in the materials and methods section.

2.      Some characterizations, i.e., XRD, BET, etc., may be added to explain the study in detail.

3.      The methods reported for electrochemical measurements are wrong. The specific capacitance is calculated by dividing the area enclosed (IxV) by the CV curve by surface area and scan rate; similarly, the energy density cannot be divided by surface area.

4.      How was the surface area of the electrode calculated in this study?

Reviewer 3 Report

In this article, an on-chip interdigitated all solid-state supercapacitor was fabricated based on porous gold electrodes and a PVA/KOH quasi-solid electrolyte. The fabrication of interdigitated porous gold electrode is performed using an original bottom-up ap-proach. Porosity is made by a templating method using a wet chemistry process followed by microfabrication techniques. The integrated micro-supercapacitor was characterized in terms of electrochemical properties and aging behavior. There are questions:

1. “We” should be avoided in the article, please state in the passive voice.

2. Please check carefully whether the space is added between number and symbol, or between word and symbol in figures, table and text or not. Such as “2.5wt. %, 12.5wt. %,  8wt. %” should be “2.5 wt.%, 12.5 wt.%, 8wt.%”;

3. Milliliter “ml” should be “mL”.

4. Please carefully check whether the superscript and subscript are correct, such as “Au/ZrO2” in conclusions, “2” should be subscript; “500 mV.s-1” in conclusions, “-1” should be superscript, etc.

5. “figure Figure 4(a)” in 3.3 should be “Figure 4(a)”.

6. Figure 4(a) and Figure 4(b) are not clear. It doesn't show what the authors are trying to say.

Reviewer 4 Report

In this manuscript, authors demonstrated the fabrication of an on-chip interdigitated all solid-state supercapacitor, based on porous gold electrodes and a PVA/KOH quasi-solid electrolyte. The manuscript is also well-organized. However, there are still some issues to be addressed. A moderate revision is suggested before its acceptance.

1. Figure1., the fabrication process, is not clearly marked in the figure.

2. The fabrication of the m-SC should be reported in the section 2, as in the last paragraph of the introduction.

3. When demonstrating “Active materials for m-SCs electrodes are mainly made for metal oxides, nanostructured carbon or conductive polymers”, some recent, relevant and important review articles should be cited: Nanocellulose and Its Derived Composite Electrodes toward Supercapacitors: Fabrication, Properties, and Challenges; Recent progress in carbon-based materials for supercapacitor electrodes: a review; Design and fabrication of conductive polymer hydrogels and their applications in flexible supercapacitors; Emergence of melanin-inspired supercapacitors; etc.

4. In 3.4 Electrochemical characterizations of the devices, the description, The shape of curves is almost square., is not accurate.

5. There is a border in Figure 6(a), which is in the different format with other figures.

6. In figure 7, the inflection point at 6000 cycles is unexplained. In addition, the insert image in Fig. 7 should be modified with better resolution.

7. How about the cost of using Au? Why not us other cheaper metal?

8. There are still some minor typos and grammar issues, for example, in 3.3 “figure Figure 4(a)”. Authors should carefully recheck the whole manuscript.

9. Authors spent too much to introduce their results. More comparison to different kinds of electrodes materials for supercapacitors should be performed by following supporting articles: Chitin derived nitrogen-doped porous carbons with ultrahigh specific surface area and tailored hierarchical porosity for high performance supercapacitors; Facile Electrodeposition of NiCo2O4 Nanosheets on Porous Carbonized Wood for Wood-Derived Asymmetric Supercapacitors; Camellia PollenDerived Carbon with Controllable N Content for HighPerformance Supercapacitors by Ammonium Chloride Activation and Dual NDoping; ZnCl 2 regulated flax-based porous carbon fibers for supercapacitors with good cycling stability; Synthetic melanin facilitates MnO supercapacitors with high specific capacitance and wide operation potential window; etc.

10. Authors cited too few references to support the reasoning, more works should be cited.
